# Effects of Socioeconomic Environment on Physical Activity Levels and Sleep Quality in Basque Schoolchildren

**DOI:** 10.3390/children10030551

**Published:** 2023-03-15

**Authors:** Arkaitz Larrinaga-Undabarrena, Xabier Río, Iker Sáez, Aitor Martinez Aguirre-Betolaza, Neritzel Albisua, Gorka Martínez de Lahidalga Aguirre, José Ramón Sánchez Isla, Mikel Urbano, Myriam Guerra-Balic, Juan Ramón Fernández, Aitor Coca

**Affiliations:** 1Department of Physical Activity and Sport Science, Faculty of Education and Sport, University of Deusto, 48007 Bilbao, Spain; 2Department of Physical Activity and Health, Osasuna Mugimendua Kontrola S.L. Mugikon, 48450 Etxebarri, Spain; 3Faculty of Humanities and Education Science, Mondragon University, 20540 Eskoriatza, Spain; 4Athlon Cooperative Society, 20500 Arrasate, Spain; 5Faculty of Health Sciences, University of Deusto, 48007 Bilbao, Spain; 6Faculty of Psychology, Education and Sport Sciences—Blanquerna, University Ramon Llull, 08022 Barcelona, Spain; 7Public College of Sports Teachings, Kirolene, Basque Government, 48200 Durango, Spain; 8Department of Physical Activity and Sports Sciences, Faculty of Health Sciences, Euneiz University, 01013 Vitoria-Gasteiz, Spain

**Keywords:** physical activity, sleep, public and private centre, schoolchildren

## Abstract

The socioeconomic and built environment of an area are interrelated with health data and have a direct influence on children’s development. There are facilitators and barriers for schools to promote physical activity depending on the socioeconomic status of the school. The aim of this study was to analyse the relationship between physical activity and sleep and the socioeconomic level of children in the Basque Country. The sample consisted of 1139 schoolchildren between the ages of six and seventeen (566 boys and 573 girls) from 75 schools (43 public and 32 private). Differences between groups were compared using the Mann–Whitney U test (two samples), Kruskal–Wallis one-factor ANOVA (k samples), and Spearman’s Rho correlation. There are sex differences in light (200.8 ± 62.5 vs. 215.9 ± 54.7) and moderate (69.0 ± 34.3 vs. 79.9 ± 32.1) physical activity in favour of the female group of higher socioeconomic status compared to male group of higher socioeconomic status. In the case of vigorous physical activity, the female group performed less than the male group across all socioeconomic statuses, which was statistically significant in the groups of high socioeconomic status (11.6 ± 9.3 vs. 6.9 ± 5.7) in group 2 and medium socioeconomic status (11.1 ± 9.3 vs. 7.7 ± 6.1) in group 3. There is an inverse relationship between sedentary behaviour and BMI, total bed time, total sleep time, and night-time awakenings. There is also an inverse relationship between all levels of physical activity performed with respect to BMI and total sleep efficiency. These data point towards notable inequalities in physical activity and daily sleep in Basque schoolchildren, which in turn may be marginalised in our current school system due to the effects of the socioeconomic environment.

## 1. Introduction

The health of individuals and populations is determined by a set of factors that go far beyond those of a biomedical nature, i.e., the genetic load and biological characteristics of individuals and their interaction with their environment [1]. On the contrary, it is living and working conditions, and other more structural factors such as the characteristics of the social, economic, and political context, which we call social determinants of health, that have a more direct influence on the healths of individuals and populations in our societies [2,3].

The fact that these social determinants of health are unequally distributed in the population generates social inequalities in health, i.e., systematic differences in health between people of different social levels, sex, ethnicity or place of residence, among other factors, meaning that the most disadvantaged groups systematically present a worse state of health. Therefore, equity in health is conditioned by the so-called structural determinants and intermediate determinants [1]. The former include aspects related to the socioeconomic and political context, which refer to the characteristics of the social structure of a society. These contextual factors exert a strong influence on patterns of social stratification, which determine the social position that people occupy in society according to their socioeconomic status, sex, level of education, place of birth, and other dimensions [3,4]. This unequal social position in turn generates inequalities in the distribution of intermediate determinants, which include living and working conditions, psychosocial factors such as the extent and quality of social networks, stress and perceived control over one’s life, and health-related behaviour such as alcohol consumption, smoking, diet, and physical activity (PA) [5].

The practice of PA and exercise has multiple health benefits: it is associated with reduced all-cause mortality, improved health-related quality of life (vitality, general health, and mental health) [6], and a reduced risk of adult-onset diabetes, obesity, osteoporosis, some cancers, and cardiovascular disease [7,8,9,10]. Accessibility and shorter distance to environments associated with physical exercise are known to increase the frequency of exercise [11,12,13,14,15]. According to ecological models, the built environment exerts a crucial influence on PA behaviour [16]. This is supported by several systematic reviews showing that people living in walkable, safer, and greener neighbourhoods tend to have higher levels of PA [17,18,19,20,21,22]. One’s socio-economic position, at the individual or area level, and the built environment are interrelated, and the path for mediation and moderation should be considered when related to health outcomes [23].

The influence of the proximity of health-related facilities on the practice of physical exercise can occur in two ways; the first has to do with the influence of seeing people in the vicinity performing physical exercise, which translates into its perception as a positive social norm [14]. The second is that one of the reasons most frequently given for abandoning physical exercise is related to the distance to the environment in which it is practiced, so the proximity of these infrastructures can eliminate the physical and psychological barriers and increase the frequency of physical exercise [14].

The importance of sleep for health at all stages of life has been widely demonstrated, but it is of vital importance in childhood and adolescence [24,25]. A deficiency in the quality of sleep can affect school performance and one’s appropriate PA levels, result in health disorders, etc. [26]. Different studies have considered the hypothesis of bidirectionality between PA and sleep, more specifically, that appropriate levels of PA are associated with better sleep efficiency and thus, deficiencies in sleep lead to lower PA levels [27]. Different studies [28,29,30] report the difficulties of children and adolescents and how it is necessary to deepen the study of these variables due to the importance of childhood and adolescence as stages of development and adherence to behaviours that will lead to future healthy adults.

The socioeconomic variable has a direct influence on children’s development [31,32,33,34]. Socioeconomic inequalities may cause altered sleep and PA patterns [35], as there is evidence linking family socioeconomic status with sleep quality [36] and PA performance [35,37]. A low socioeconomic status affects children’s health and can have lifelong consequences, in both early and later life [38,39]. In addition, physical activity is not equally distributed across social classes, with the most disadvantaged performing less exercise during both adolescence [40] and adulthood [41].

Current policies are clear about the importance of sport and PA: sport is healthy and there is still much to be achieve in the area of healthy lifestyles [42]; however, in low-, middle-, and high-income countries, PA levels are still insufficient [31,43]. Moreover, few low- and middle-income countries have PA policies, while policies have been well developed in many high-income countries, although often with very limited implementation [44]. In addition, for many families today, PA involves a financial cost and family time that not everyone has the resources to afford [45]. In this situation, it is easier to choose other solutions to introduce PA, such as free school sports and informal PA in public spaces such as parks [46,47]. However, these children are more likely to engage in sedentary activities, such as playing video games instead of PA, which can lead to poor physical fitness in adolescence [48]. Contemporary global socio-cultural and physical environments are generally not conducive to high levels of regular PA among children and adolescents [31]. Instead, these have produced abnormal activity habits and social norms, limited access to the means of fulfilling their basic biological needs, and denied children the human right to physically active games [49]. In this regard, children of lower socioeconomic status residing in different European countries are less likely to participate in sports clubs, more likely to spend more than two hours a day in front of a screen [47], and less likely to spend more than two hours in an open environment on weekends compared to those of higher socioeconomic status [50]. The environmental changes that have reduced PA among children and adolescents have also led to biological changes, including reduced motor competence, reduced physical fitness, and high body fatness, even among those who are not overweight or obese as defined by their body mass index. In turn, these biological changes have further reduced the PA by producing feedback loops that amplify adverse environmental impacts on PA [51,52].

In the United States, there is a direct relationship between family wealth and the ability to participate in organised sports [29]; therefore, the children of families of higher socioeconomic status are more likely to meet the recommended levels of PA and sports participation [39]. Families of a higher socioeconomic status usually have more financial resources for their children to engage in extracurricular activities and may know more about the importance of the impact of PA on health. Therefore, it is easier to encourage these parents to actively participate in sports clubs [45,47]. In summary, children of high socioeconomic status have higher levels of PA [33,39,48]. In addition, children’s access to green spaces in urban areas is closely related to their physical and psychological well-being [53].

The size of the spaces available for play in the school environment directly influences the PA to be practiced. Schools of higher socioeconomic status will have more learning materials and equipment at their disposal, such as balls, ropes, or other materials for children to play with. At the same time, this influences the PA to be practiced at recess. Sports fields, green areas, trees, games, concrete, and shaded areas make it easier to perform PA, and in less discriminatory centres, the children obtained higher caloric expenditure in the games [54]. In order to have the capacity to improve the PA levels of children, centres must give importance to the provision of resources for PA, contributing to the creation of a culture of PA [55]. In relation to this, there are facilitators and barriers for schools to promote PA and raise the levels of PA among children, with some predominant facilitators or barriers depending on the socioeconomic level of the school [55,56]. In primary schools of high socioeconomic level, the lack of barriers related to the curriculum, teacher proficiency, and the intrinsic factors of individual pupils mean that schoolchildren have higher levels of PA [56]. At the compulsory secondary education stage, on the other hand, schools of low socioeconomic level have more barriers than high socioeconomic schools, such as those related to school policy, environment, and individual students’ intrinsic factors [57]. This leads to the fact that socioeconomic disadvantages among girls predict negative knowledge and achievement outcomes [38,58].

For all these reasons, the aim of this study was to analyse the relationship between PA and sleep and the socioeconomic level of girls and boys between 6 and 17 years of age in the Basque Country.

## 2. Materials and Methods

### 2.1. Subjects and Design

A cross-sectional observational study was carried out. Participants were selected by non-probabilistic convenience sampling across all schools in the Basque Country. The study sample consisted of 1139 schoolchildren between the ages of six and seventeen (566 boys and 573 girls) from 75 schools (43 public and 32 private) that gave their definitive approval to the study. A proportional and random stratification according to historical territory (Araba, Bizkaia, and Gipuzkoa), sex, age (primary education from 6 to 12 years and secondary education from 12 to 17 years) and ownership of the school (public or private) was taken into account. The public schools were 100% publicly funded while the private schools analysed in this study were co-financed, wherein the education was free but other services such as canteens, transport, materials, and other school activities were not.

The qualitative variables of the study were sex, educational stage (primary education and compulsory secondary education), school, and socioeconomic level (SEP). On the other hand, the quantitative variables were body mass index (BMI), PA levels (light, moderate, vigorous, and MVPA), sedentary behaviour (min), total time in bed (min), total sleep time (min), night-time awakenings (min), and sleep efficiency (%).

### 2.2. Instruments

The ActiGraph WGT3X-BT accelerometer (manufacturer ActiGraph, 49 East Chase St. Pensacola, FL, USA) was used to collect data related to PA levels as well as sleep parameters. The participants wore the accelerometer for seven consecutive days including a weekend. The device was worn on the non-dominant hand. Recordings were considered valid with a minimum daily exposure of 10 h for at least 3 days, among which at least 2 have to be working days and one on the weekend. In addition, it was requested that the accelerometer be removed during bathing, showering, and/or other water-based activities. They were collected based on the validity and reliability of previous studies [59,60,61,62].

### 2.3. Procedure

In order to carry out the research, approval was requested from the Basque Medicines Research Ethics Committee (Basque Government Department of Health) in accordance with the Law 14/2007 on biomedical research [63], the ethical principles of the Helsinki Declaration of 2013 [64], and other ethical principles and applicable legislation in the report of the Basque Medicines Research Ethics Committee (CEIm-E) of the Basque Government Department of Health with internal code PI2020011. Likewise, the current regulations on personal data protection were respected: namely (EU) Regulation 2016/679 of 27 April 2016 (GDPR) [65], Organic Law 3/2018 of 5 December on Personal Data Protection and guarantee of digital rights (ES) [66], and Royal Decree (ES) 1720/2007 of 21 December [67]. In all these documents and permits, it was taken into account that the study included school-aged children.

Following the approval of the project, the Department of Education of the Basque Government sent an e-mail to all schools in the Basque Country (Figure 1). Subsequently, positive responses were collected from the schools interested in participating and meetings were held with the school’s management teams as well as with the physical education teachers through whom the families were provided with the information and documentation of this study. Among all the families that were willing to participate, participants were selected by means of a draw among those who met the selection criteria established for this study. Then, all the children’s legal guardians signed the informed consent form and the pupils themselves signed their informed consent. Once the participants by school, grade, and sex had been confirmed, a timetable was established for placing and removing the accelerometers.

There is a proportional and random stratification according to the historical territory and county, age, sex, educational network (public or private), and SEP index (socioeconomic level based on the deprivation index in the census section, which makes it possible to identify sections with socioeconomic conditions), together with inclusion and exclusion criteria that can be seen in Table 1.

In this study, for the determination of socioeconomic status, hereafter SEP, the MEDEA classification was used [68], which is a socioeconomic level that calculates the average per capita income of people living in the district in which the school is located, divided into five groups (see Table 2).

### 2.4. Statistical Analysis

For the outcome variables, descriptive statistics were used, reporting the level of significance for the main group (between participants). To avoid a type I error, a post hoc analysis was performed when the interaction effect was found to be significant. Values will be expressed as the mean (SD).

Statistical analysis was performed with SPSS software (version 28.0.1.0; IBM Corp; Armonk, New York, NY, USA). Values of *p* < 0.05 were considered statistically significant. First, the Kolmogorov–Smirnov test was used to assess the normality of the distribution and Levene’s test to observe the homogeneity of variances, as well as an analysis of the descriptive variables studied (means, standard deviation, etc.). None of the variables studied met the above requirements, so the differences between the groups were compared using the non-parametric Mann–Whitney U test (2 samples) and the Kruskal–Wallis one-factor ANOVA (k samples). After a significant Kruskal–Wallis H test, a Dunn–Bonferroni test was used for pairwise post hoc comparisons. Correlation between the variables was estimated using Spearman’s Rho.

## 3. Results

The descriptive results of the sleep variables (Table 3, no statistical differences) and PA variables (Table 4) divided by sex and in each of the SEP categories are shown below.

A sex difference was observed in the performance of light and moderate PA between the SEP 1 female and male groups. In the case of vigorous PA, the female group performed less PA than males in all groups, which was a statistically significant difference between SEP groups 2 and 3.

Table 5 shows the values for the sleep parameters in the primary and secondary variables. Thus, in the primary stage, both the female and male groups have better sleep efficiency, spend less time in bed, and have fewer night-time awakenings at the two extremes of the SEP (groups 1 and 5) compared to the intermediate levels.

At the secondary stage, females spend more time in bed in SEP group 1 than in the other groups, with a significant difference compared to groups 3 and 5.

Table 6 shows the results of the PA parameters in both the primary and secondary school stages. In the primary stage, the male SEP group 5 showed greater sedentary behaviour than the rest of the groups. On the other hand, in the female group at the primary stage, although all those in the female SEP group 5 showed greater sedentary behaviour than the rest of the groups, significant differences can only be seen with groups 3 and 4. At the primary stage, intermediate male groups 2, 3, and 4 showed the highest levels of PA with respect to groups 1 and 5. 

As for the secondary stage, the schoolchildren who showed the greatest sedentary behaviour, amongst both males and females, belong to group 1. In addition, the female SEP group 5 of this stage was that with the least moderate PA and MVPA compared to the rest of the groups.

Table 7 and Table 8 show the results with respect to the school ownership variable. Specifically, Table 7 shows the results for BMI and sleep quality. Among the public school students, males in SEP group 4 have the highest BMI among the 5 groups. In addition, males in SEP group 1 have the highest sleep efficiency, with fewer night-time awakenings. On the other hand, regardless of sex, males in SEP group 3 spend the most time in bed, sleep the longest, and have the most night-time awakenings. 

Among private school children, females in SEP group 2 show lower BMI values compared to the other groups. In addition, males in SEP group 1 show lower values for time spent in bed and time spent asleep compared to groups 2 and 4.

Table 8 shows the daily PA values within the school ownership variable (public and private) analysed by sex in terms of the PA variables. Male and female schoolchildren in public schools, specifically those belonging to SEP group 4, showed the highest levels of sedentary behaviour with respect to the rest of the groups, which was statistically significant in females with respect to the rest of the groups. These values of sedentary behaviour are reflected in the higher BMI (see Table 7) in both male and female schoolchildren, although the latter is not statistically significant. In addition, the male SEP group 4 of the public centres were those performing the least vigorous PA. 

In private centres, it is both male and female SEP groups 1 and 5 who show the highest values of sedentary behaviour. On the other hand, those in SEP group 2 show the lowest values of sedentary behaviour and the highest values of moderate, vigorous, and MVPA compared to the rest of the groups. The female schoolchildren in SEP group 2 in these private centres also have the highest levels of light, moderate, and MVPA PA compared to the rest of the groups. Conversely, in females from public centres, the values of light, moderate, vigorous, and MVPA are lower in SEP group 4 than in groups 1 and 3.

Table 9 shows the correlation values of the total sample between all the variables analysed. Thus, there is an inverse relationship between sedentary behaviour and BMI, total bed time, total sleep time, and WASO. There is also an inverse relationship between all levels of PA performed with respect to BMI and total sleep efficiency.

## 4. Discussion

The aim of this study was to analyse the relationship between PA and sleep with the socioeconomic level of girls and boys between 6 and 17 years of age in the Basque Country. A meta-analysis on the sleep–obesity relationship in children and adolescents [69] emphasised a research recommendation on the interaction of demographic factors with sleep and obesity. MVPA, sedentary behaviour and demographic information, such as sex, age, and parental education level, were included as covariates because they were reported to influence the weight status and sleep of children and adolescents [70,71]. A short sleep duration is associated with an increased risk of overweight/obesity in children and adolescents in a study performed in China, independently of sleep quality. This relationship is significant for children rather than adolescents. Short sleep duration and sleep quality were significantly associated with overweight/obesity in girls, but not in boys in the same study [72,73]. Considering sleep quality, some studies have found that children and adolescents with poor sleep quality are more likely to gain weight [70,74], while other studies found no significant relationship between the two variables [75,76]. In our case, children have better sleep efficiency, spend less time in bed, sleep less, and have fewer night-time awakenings, which was true for both males and females of primary school age in the SEP extremes (groups 1 and 5) compared to schoolchildren in groups 2, 3, and 4. Females who spend more time in bed are those in a high SEP (group 1), which was a significant difference compared to females in the lowest SEP (group 5).

Schoolchildren with lower SEP have a higher PA than their peers with higher SEP (particularly because of their greater participation in more active transportation, more household chores, and more work-related activities), whilst the latter participated more frequently in organised sports and formal activities [77]. However, for the most part, a high SEP is associated with a considerable increase in the frequency of PA [33,77,78,79,80,81,82]. In primary school, children from middle- and high-income families are approximately three times more likely to meet PA recommendations compared to children from low SEP families. For secondary school students, children from middle-income families are twice as likely and those from high-income families are more than three times as likely to ever participate in sports compared to children from low SEP families [39]. This relationship may be due to both the cost of access to sports practices being a barrier for families with lower purchasing power [83]. However, lower wealth is also associated with children’s participation in optimal amounts of health-promoting PA [39].

The research evidence suggests that PA behaviours are socioeconomically shaped, as children with a low socioeconomic status spend less time being physically active during leisure time and engage in less vigorous intensity activities compared to their peers of high economic status [84,85]. In our case, in public centres, males in SEP 4 were those performing the least vigorous PA, and females had the lowest PA values in light, moderate, vigorous, and MVPA, compared to groups 1 and 3. Furthermore, in private centres, both males and females in SEP 2 are those showing the highest PA values in moderate, vigorous, and MVPA. High-income children engage in significantly more vigorous activity than low- and middle-income children do [85]. These domain-specific and intensity-specific differences are important, as vigorous PA is considered to elicit stronger health benefits compared to lower intensity PA [86]. Among most northern, eastern, and southern European countries, children with low parental education played actively/vigorously for longer periods of time [33]. Meanwhile, the opposite situation emerged among Central Asian countries. An inverse socioeconomic gradient also emerges in relation to the practice of sports, with lower SEP children being less involved in these activities. On average, 70.9% of children from families with low parental education level spend less than 2 h/week doing sport compared to 38.2% of children with high parental education level [45]. The higher the level of education, the higher the educational climate and the higher the likelihood of performing PA [87].

In Canada, SEP at the area level was not related to step count or the amount of time children spent performing MVPA [50]. However, another study found that children in areas with higher SEP were more likely to comply with the daily step count recommendations [81]. In our case, in the primary stage, male schoolchildren with a lower SEP showed greater sedentary behaviour compared to the rest of the categories of the index. In contrast, in females of lower SEP in the primary stage, even though they showed greater sedentary behaviour than categories 3 and 4, there were no significant differences with the highest SEP categories (SEP 1 and 2), coinciding with another study that found that the amount of time spent performing MVPA (moderate to vigorous PA) was not statistically different for children from low-, medium-, or high-SEP households [88].

In Australia, children studying in schools of high socioeconomic status are more likely to meet the recommended PA levels as well as healthy cardiorespiratory fitness levels compared to children studying in schools with a low socioeconomic status. In the secondary stage, however, there were no significant differences [39], unlike in the present study, where, in the secondary stage, the schoolchildren who showed greater sedentary behaviour, in both males and females, belonged to the highest SEP (SEP 1). In addition, females with the lowest SEP (SEP 5) at this stage are those with the least moderate PA and MVPA. In turn, most of the barriers are related to curriculum, teaching and school policy, and environment [83]. In the present study, male and female public school children, specifically those belonging to low SEPs, show the highest levels of sedentary behaviour compared to other deprivation levels. In addition, low-SEP males in public schools are the least likely to engage in vigorous PA. Consistently with the results, which indicate that children from low-SEP families may be more likely to engage in sedentary activities, such as screen use, rather than recreational and physical activities, this in turn may lead to poor physical fitness in adolescence [48]. For families of low SEP, the lack of resources to enrol their children in a sporting activity (e.g., football, judo, gymnastics, jazz, ballet, tennis, etc.) could play an important role [47].

Young people of higher SEP are those who dedicate more time to sedentary behaviour, while those of families with lower incomes practice PA in a much less relevant way [89], coinciding with the results obtained in the private centres in our study, that, in an almost opposite way to the results obtained in the public centres, male and female schoolchildren and females with higher SEP (SEP 1) are those that reflect a higher level of sedentary behaviour. However, in another study, there was no clear pattern in terms of socioeconomic status with respect to inactivity, i.e., the subjects analysed all showed practically the same data regardless of whether they were of high or low SEP [82].

Children of low SEP generally have a higher BMI, more behavioural difficulties, report a lower quality of life, and experience more critical life events than children with a higher SEP [90], as shown in the males and females of the public school of the present study, specifically those belonging to low SEP, the highest levels of sedentary behaviour with respect to the rest of the deprivation levels. Values of sedentary behaviour are reflected in the higher BMI in males and in females, although the latter is not statistically significant. Schoolchildren of a lower SEP tend to be more obese (12.6% compared to 8% of more advantaged children) [91].

Similarly to a representative sample of Swedish adolescents [92], females of high SEP (SEP 2) in private schools have the highest levels of light, moderate, and MVPA compared to the other categories of the index. Conversely, in females in public schools, light, moderate PA, vigorous PA, and MVPA values are lower in low SEP (SEP 4) than in higher and medium SEP (SEP 1 and 3) groups. 

Children of low socioeconomic backgrounds tend to be more prevalent in groups that combine multiple unhealthy lifestyles. Thus, children whose mother received a low educational level or children from a low-income household have been classified in groups that are physically inactive and engage in significant screen time [47,93].

The limitations of the present study have been associated with the difficulties of involving educational centres and the agents that form part of the schools; teachers, parents, and students. In addition, it was requested that the accelerometers be removed from the wrists when engaging in aquatic activities such as baths or showers, so there is very interesting information that is not collected in the practitioners of aquatic activities. With regard to future lines of research, on the one hand, the relative percentile values of the volume of PA could be analysed as a function of variables such as sex or age. On the other hand, it could be used to study the variables related to sleep quality. It would be highly intriguing to conduct a longitudinal study that researches the temporal changes in PA and sleep quality among child and adolescents in Basque Country, as well as the correlations between all these variables.

## 5. Conclusions

The results of our study show a positive trend in both sedentary behaviour and sleep efficiency in Basque schoolchildren of intermediate SEP with respect to the extremes. In addition, it was observed that females performed less high-intensity PA than males in all SEP groups. 

Likewise, more sedentary behaviour was associated with less time spent in bed and poorer sleep efficiency. Similarly, more PA, regardless of its intensity, was associated with better body composition. 

These data point to notable inequalities in PA and daily sleep among Basque schoolchildren which, in turn, may be marginalised in our current school system due to the effects of the socioeconomic environment. It is therefore interesting to address future strategies in a cross-cutting manner so that schoolchildren extend their hours of movement both in and out of the classroom.

## Figures and Tables

**Figure 1 children-10-00551-f001:**
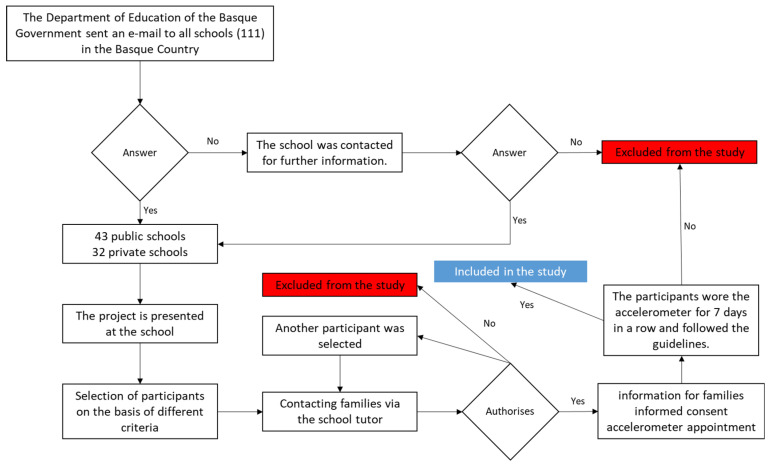
Participants reclusion flow chart.

**Table 1 children-10-00551-t001:** Inclusion and exclusion criteria.

Inclusion criteria	Belong to a school centre adhering to for program at primary or secondary school levels.
	The parents or legal guardians of the student must complete the informed consent form.
Exclusion criteria	Failure to comply with any of the following conditions: negative consent by the person legally responsible for the student, failure to complete the qualitative record (daily), incorrect use of the accelerometer.
	Disability or inability to complete the qualitative record or correctly use the accelerometer according to protocol. In the case of doubt regarding any participant, the teaching staff of the school will be involved in the decision to include or exclude.

**Table 2 children-10-00551-t002:** SEP term assigned to each group.

SEP	Assigned Term
SEP 1	Group 1
SEP 2	Group 2
SEP 3	Group 3
SEP 4	Group 4
SEP 5	Group 5

SEP 1, highest economic position group; and SEP 5, lowest economic position group.

**Table 3 children-10-00551-t003:** Results of the daily sleep quality analysis in each SEP level divided by sex.

SEP	Sex	Sleep Efficiency %	Total Bed Time (min)	Total Sleep Time (min)	WASO (min)
1	M (*n* = 120)	87.5 ± 6.2	468.6 ± 82.2	409.1 ± 75.5	56.1 ± 33.0
	F (*n* = 122)	87.2 ± 5.7	462.5 ± 76.0	401.3 ± 66.9	57.7 ± 28.9
2	M (*n* = 113)	86.2 ± 6.6	494.3 ± 78.3	425.4 ± 67.2	65.2 ± 37.8
	F (*n* = 120)	87.2 ± 5.8	481.6 ± 80.6	417.5 ± 61.7	59.7 ± 30.6
3	M (*n* = 162)	86.5 ± 6.4	491.5 ± 84.4	421.7 ± 64.1	65.5 ± 35.0
	F (*n* = 147)	86.6 ± 5.7	495.3 ± 85.2	426.8 ± 69.6	63.3 ± 29.7
4	M (*n* = 105)	86.3 ± 6.6	489.5 ± 85.1	419.9 ± 71.9	61.8 ± 33.8
	F (*n* = 111)	86.6 ± 7.0	487.9 ± 82.1	419.1 ± 65.6	60.4 ± 35.4
5	M (*n* = 66)	87.1 ± 6.5	459.5 ± 77.4	398.5 ± 64.8	54.4 ± 32.4
	F (*n* = 73)	88.6 ± 5.5	457.8 ± 91.0	404.9 ± 80.9	55.0 ± 28.2

Data are presented as mean ± SD. M: male; F: female; WASO: wake after sleep onset.

**Table 4 children-10-00551-t004:** Results of the daily sleep quality analysis in each SEP level divided by sex.

SEP	Sex	Sedentary (min)	Light PA (min)	Moderate PA (min)	Vigorous PA (min)	MVPA
1	M (*n* = 120)	625.6 ± 202.5	200.8 ± 62.5	69.0 ± 34.3	8.7 ± 7.5	77.6 ± 40.5
	F (*n* = 122)	614.8 ± 171.7	215.9 ± 54.7 *	79.9 ± 32.1 *	6.3 ± 4.7	86.2 ± 35.7
2	M (*n* = 113)	573.4 ± 145.0	215.2 ± 54.2	81.1 ± 36.5	11.6 ± 9.3 ***	92.7 ± 44.1
	F (*n* = 120)	541.6 ± 171.4	216.0 ± 72.2	81.4 ± 36.6	6.9 ± 5.7	88.3 ± 40.6
3	M (*n* = 162)	551.4 ± 175.6	212.7 ± 66.4	80.7 ± 40.4	11.1 ± 9.3 **	91.7 ± 48.2
	F (*n* = 147)	542.2 ± 167.2	223.2 ± 63.1	85.8 ± 33.5	7.7 ± 6.1	93.4 ± 38.2
4	M (*n* = 105)	556.7 ± 214.3	202.7 ± 76.8	71.8 ± 41.0	9.1 ± 9.3	81.0 ± 49.0
	F (*n* = 111)	600.0 ± 199.2	210.7 ± 62.3	79.8 ± 36.9	6.7 ± 6.9	86.5 ± 42.3
5	M (*n* = 66)	583.1 ± 227.1	201.2 ± 73.9	69.9 ± 37.4	8.1 ± 8.0	78.0 ± 44.2
	F (*n* = 73)	563.1 ± 201.7	208.3 ± 78.3	77.4 ± 39.5	6.8 ± 5.2	84.3 ± 43.8

Data are presented as mean ± SD. M: male; F: female; PA: physical activity; MVPA: moderate to vigorous physical activity. * Statistical inter group differences between males and women. * *p* ≤ 0.05; ** *p* ≤ 0.01; *** *p* ≤ 0.001.

**Table 5 children-10-00551-t005:** Results of the daily sleep quality analysis in each SEP level divided by sex and school level group.

School Group	SEP	Sex	BMI (kg/m^2^)	Sleep Efficiency (%)	Total Bed Time (min)	Total Sleep Time (min)	WASO (min)
Primary 6–12	1	M (*n* = 59)	18.08 ± 4.03	88.13 ± 5.84 ^2,3,4^ 87.82 ± 4.99 ^3,4^	466.48 ± 88.8 ^2,3,4^ 448.68 ± 82.72 ^2,3,4^	410.75 ± 79.43 ^2,3,4^ 392.60 ± 75.04 ^2,3,4^	52.81 ± 32.53 ^2,3,4^ 54.16 ± 24.32 ^2,3,4^
		F (*n* = 66)	17.79 ± 2.88				
	2	M (*n* = 62)	17.51 ± 2.55	85.72 ± 6.89 ^1^ 85.52 ± 6.31 ^5^	521.73 ± 81.15 ^1,5^ 508.66 ± 88.08 ^1,5^	445.24 ± 67.76 ^1,5^ 432.07 ± 65.49 ^1^	73.38 ± 42.63 ^1,5^ 67.98 ± 32.99 ^1^
		F (*n* = 58)	17.05 ± 2.03				
	3	M (*n* = 104)	17.36 ± 2.64	84.98 ± 6.02 ^1,5^ 85.76 ± 4.99 ^1,5^	515.10 ± 86.01 ^1,5^ 518.15 ± 92.22 ^1,5^	434.17 ± 63.51 ^1,5^ 442.15 ± 74.84 ^1,5^	74.50 ± 35.22 ^1,5^ 72.69 ± 28.77 ^1,5^
		F (*n* = 95)	17.79 ± 2.80				
	4	M (*n* = 36)	18.55 ± 3.73	84.46 ± 6.08 ^1,5^ 84.16 ± 7.38 ^1,5^	537.30 ± 75.49 ^1,5^ 527.41 ± 86.25 ^1,5^	450.46 ± 50.27 ^1,5^ 438.11 ± 55.17 ^1^	74.87 ± 32.82 ^1,5^ 76.52 ± 40.96 ^1,5^
		F (*n* = 38)	16.80 ± 2.01				
	5	M (*n* = 51)	17.52 ± 2.63	86.65 ± 6.28 ^3,4^ 88.51 ± 3.80 ^2,3,4^	467.38 ± 80.33 ^2,3,4^ 466.23 ± 95.62 ^2,3,4^	403.54 ± 68.32 ^2,3,4^ 411.12 ± 80.76 ^3^	57.38 ± 33.49 ^2,3,4^ 57.30 ± 25.19 ^3,4^
		F (*n* = 54)	17.24 ± 2.79				
Secondary 12–17	1	M (*n* = 59)	20.23 ± 2.72	86.67 ± 6.43	469.80 ± 74.35	405.34 ± 68.82	60.44 ± 33.24
		F (*n* = 56)	19.70 ± 2.46 ^1,4^	86.49 ± 6.54	478.74 ± 64.29 ^3,5^	411.49 ± 54.77	61.78 ± 33.34
	2	M (*n* = 51)	20.22 ± 3.83	86.88 ± 6.12	460.97 ± 60.20	401.21 ± 58.65	55.25 ± 28.32
		F (*n* = 62)	20.31 ± 2.93^4^	88.77 ± 4.90	456.34 ± 63.76	403.77 ± 54.97	56.00 ± 26.21
	3	M (*n* = 58)	20.52 ± 3.05	89.10 ± 6.14	449.25 ± 62.44	399.29 ± 59.36	49.23 ± 28.25
		F (*n* = 52)	20.54 ± 2.90^4^	88.12 ± 6.66	453.62 ± 48.36 ^1^	398.83 ± 47.92	46.02 ± 23.01
	4	M (*n* = 69)	20.90 ± 2.51	87.19 ± 6.66	464.49 ± 79.40	403.94 ± 76.51	55.02 ± 32.45
		F (*n* = 73)	21.58 ± 3.39 ^1,2,3,5^	87.88 ± 6.47	467.30 ± 71.78 ^5^	409.26 ± 68.69	52.05 ± 29.12
	5	M (*n* = 15)	27.33 ± 31.26	88.71 ± 7.04	432.66 ± 61.12	381.40 ± 49.06	44.43 ± 26.88
		F (*n* = 19)	21.03 ± 7.31 ^4^	88.91 ± 8.73	433.89 ± 73.23 ^1,4^	387.05 ± 80.85	48.45 ± 35.51

Data are presented as mean ± SD. M: male; F: female; BMI: body mass index; WASO: wake after sleep onset. ^1,2,3,4,5^ Statistical inter-group differences between SEPs.

**Table 6 children-10-00551-t006:** Results of the daily PA analysis in each SEP level divided by sex and school level group.

School Group	SEP	Sex	Sedentary (min)	Light PA (min)	Moderate PA (min)	Vigorous PA (min)	MVPA (min)
Primary 6–12	1	M (*n* = 59)	573.71 ± 162.45 ^3,4,5^	220.16 ± 55.90 ^2,3,4^ 237.56 ± 53.97	88.33 ± 30.36 ^2,3,4^ 95.76 ± 31.10	12.46 ± 7.00 ^2,3,4,5^ 8.56 ± 4.83	100.80 ± 35.39 ^2,3,4^ 104.33 ± 34.52
		F (*n* = 66)	549.66 ± 152.63 ^3,4^				
	2	M (*n* = 62)	558.03 ± 160.84 ^3,4,5^	234.43 ± 53.10 ^1,5^ 248.40 ± 55.25	101.19 ± 30.90 ^1,5^ 99.72 ± 30.67	15.47 ± 8.78 ^1,5^ 9.07 ± 5.09	116.66 ± 37.78 ^1,5^ 108.79 ± 34.18
		F (*n* = 58)	543.62 ± 145.12 ^3,4^				
	3	M (*n* = 104)	514.76 ± 141.80 ^1,2,5^	233.72 ± 58.32 ^1,5^ 231.03 ± 72.87	100.43 ± 31.68 ^1,5^ 94.45 ± 34.51	15.39 ± 8.26 ^1,5^ 9.55 ± 5.99	115.82 ± 37.60 ^1,5^ 104.00 ± 38.72
		F (*n* = 95)	481.40 ± 162.42 ^1,2,5^				
	4	M (*n* = 36)	492.76 ± 83.43 ^1,2,5^	252.89 ± 37.49 ^1,5^ 242.76 ± 55.11	105.02 ± 33.64 ^1,5^ 103.42 ± 37.64	17.12 ± 9.85 ^1,5^ 12.07 ± 8.48	122.15 ± 41.73 ^1,5^ 115.49 ± 44.08
		F (*n* = 38)	478.75 ± 126.03 ^1,2,5^				
	5	M (*n* = 51)	624.32 ± 190.64 ^1,2,3,4^	215.68 ± 59.06 ^2,3,4^	77.73 ± 36.09 ^2,3,4^	4.95 ± 5.98	87.39 ± 43.25 ^2,3,4^
		F (*n* = 54)	591.02 ± 169.85 ^3,4^	229.30 ± 63.57	89.38 ± 35.55	8.43 ± 4.93	97.81 ± 39.31
Secondary 12–17	1	M (*n* = 59)	683.77 ± 223.52 ^2,3,4,5^	182.44 ± 61.45	50.44 ± 26.49	4.95 ± 5.98	55.40 ± 30.95
		F (*n* = 56)	691.45 ± 162.05 ^2,3,4,5^	190.41 ± 43.76	61.22 ± 21.64 ^5^	3.60 ± 2.90	64.82 ± 23.085
	2	M (*n* = 51)	591.97 ± 121.95 ^1^	191.77 ± 45.98	56.55 ± 26.63	6.97 ± 7.80	63.53 ± 32.10
		F (*n* = 62)	539.66 ± 193.91 ^1,3,4^	185.59 ± 73.27	64.34 ± 33.50 ^5^	4.86 ± 5.53	69.21 ± 36.89 ^5^
	3	M (*n* = 58)	617.01 ± 209.61 ^1,5^	175.09 ± 63.85	45.21 ± 28.47	3.25 ± 4.86	48.46 ± 32.46
		F (*n* = 52)	653.17 ± 109.23 ^2,5^	208.82 ± 35.94	69.88 ± 24.81 ^5^	4.26 ± 4.84	74.15 ± 28.79 ^5^
	4	M (*n* = 69)	605.31 ± 249.52 ^1,5^	176.48 ± 79.08	54.52 ± 33.24	4.95 ± 5.59	59.47 ± 37.56
		F (*n* = 73)	662.21 ± 201.27 ^2,5^	193.93 ± 59.47	67.54 ± 30.10 ^5^	3.87 ± 3.51	71.41 ± 32.59^5^
	5	M (*n* = 15)	442.80 ± 286.98 ^1,3,4^	152.08 ± 97.56	43.26 ± 29.05	2.90 ± 3.29	46.17 ± 31.30
		F (*n* = 19)	483.87 ± 262.36	148.67 ± 87.01	43.46 ± 29.63 ^1,2,3,4^	2.31 ± 2.51	45.78 ± 31.63 ^1,2,3,4^

Data are presented as mean ± SD. M: male; F: female; WASO: wake after sleep onset. ^1,2,3,4,5^ Statistical inter group differences between SEPs.

**Table 7 children-10-00551-t007:** Results of the daily sleep quality analysis in each SEP level divided by sex and school titularity ^1,2,3,4,5^.

School Group	SEP	Sex	BMI (kg/m^2^)	Sleep Efficiency (%)	Total Bed Time (min)	Total Sleep Time (min)	WASO (min)
Public school	1	M (*n* = 34)	18.82 ± 4.23 ^4^	89.75 ± 5.13 ^2,3,4^ 87.93 ± 6.42	469.61 ± 87.17 ^3^ 439.17 ± 78.13 ^2,3,4^	424.84 ± 91.40 ^3^ 383.99 ± 75.66 ^2,3,4^	41.16 ± 20.59 ^2,3,4^ 53.61 ± 30.18 ^3^
		F (*n* = 36)	18.57 ± 2.88				
	2	M (*n* = 74)	19.01 ± 3.64 ^4^	85.26 ± 6.52 ^1^ 87.09 ± 6.05 ^5^	489.37 ± 81.47 ^3,5^ 478.33 ± 84.57 ^1,3^	415.25 ± 62.38 ^3,5^ 413.34 ± 61.03 ^1,3^	65.77 ± 36.15 ^1,3,5^ 60.29 ± 37.71 ^3,5^
		F (*n* = 84)	19.40 ± 3.10				
	3	M (*n* = 77)	18.13 ± 2.86 ^4^	84.72 ± 5.96 ^1^ 86.19 ± 5.14 ^4,5^	538.12 ± 62.38 ^1,2,4,5^ 540.52 ± 61.65 ^1,2,4,5^	453.27 ± 50.031 ^1,2,4,5^ 464.34 ± 50.28 ^1,2,4,5^	78.08 ± 32.80 ^1,2,4,5^ 71.17 ± 28.01 ^1,2,4,5^
		F (*n* = 71)	18.96 ± 3.10				
	4	M (*n* = 68)	20.26 ± 2.85 ^1,2,3,5^	86.48 ± 7.08 ^1^ 87.27 ± 7.65 ^3^	483.54 ± 90.31 ^3^ 481.13 ± 88.93 ^1,3^	416.28 ± 79.87 ^3,5^ 416.78 ± 73.65 ^1,3^	58.84 ± 35.02 ^1,3^ 56.86 ± 38.30 ^3^
		F (*n* = 69)	20.27 ± 4.00				
	5	M (*n* = 30)	18.37 ± 3.22 ^4^	86.36 ± 7.15 ^2,3^	443.10 ± 70.92 ^3,5^	381.38 ± 61.50 ^2,3,4^	47.69 ± 23.71 ^2,3^
		F (*n* = 29)	18.71 ± 3.24	89.80 ± 6.45	449.72 ± 85.72 ^3^	404.26 ± 88.86 ^3^	47.85 ± 28.27 ^2,3^
Private school	1	M (*n* = 86)	19.24 ± 3.29	86.57 ± 6.31	468.24 ± 80.73 ^2,4^	402.85 ± 67.86 ^2,4^	62.06 ± 35.08
		F (*n* = 86)	18.71 ± 2.86 ^2,5^	86.91 ± 5.48	472.23 ± 73.41	408.51 ± 61.96	59.35 ± 28.41
	2	M (*n* = 39)	18.21 ± 3.05	88.12 ± 6.29	503.68 ± 71.89 ^1,4^	444.57 ± 72.60 ^1,4^	64.11 ± 41.26
		F (*n* = 36)	17.17 ± 2.07 ^1,3,4^	87.46 ± 5.36	489.31 ± 70.82 ^3^	427.03 ± 63.01 ^3^	58.39 ± 28.32
	3	M (*n* = 85)	18.82 ± 3.42	88.02 ± 6.34	449.31 ± 79.52 ^2,4^	393.07 ± 62.20 ^2,4^	54.01 ± 33.08
		F (*n* = 76)	18.58 ± 3.15 ^2,5^	86.97 ± 6.23	453.10 ± 82.64 ^2,4^	391.78 ± 66.98	55.86 ± 29.48
	4	M (*n* = 37)	19.78 ± 3.71	85.84 ± 5.60	500.33 ± 74.60 ^1,3^	426.52 ± 54.73 ^1,3^	67.31 ± 31.07
		F (*n* = 42)	19.41 ± 3.26 ^2,5^	85.52 ± 5.66	498.96 ± 68.96 ^3^	423.00 ± 50.16 ^3^	66.30 ± 29.64
	5	M (*n* = 36)	20.90 ± 2.51	87.76 ± 5.85	473.15 ± 80.76	412.79 ± 64.79	60.05 ± 37.54
		F (*n* = 44)	17.91 ± 5.43 ^1,3,4^	87.84 ± 4.60	463.15 ± 94.86	405.24 ± 76.30	59.71 ± 27.53

Data are presented as mean ± SD. M: male; F: female; BMI: body mass index; WASO: wake after sleep onset. ^1,2,3,4,5^ Statistical inter group differences between SEPs.

**Table 8 children-10-00551-t008:** Results of the daily PA analysis in each SEP level divided by sex and type of school.

School Group	SEP	Sex	Sedentary (min)	Light PA (min)	Moderate PA (min)	Vigorous PA (min)	MVPA (min)
Public school	1	M (*n* = 34)	562.82 ± 154.26 ^5^	203.96 ± 61.60 241.30 ± 34.40 ^2,4^	76.70 ± 34.76 97.25 ± 23.52 ^2,4^	10.23 ± 6.90 ^4^ 8.43 ± 4.34 ^2,4^	86.94 ± 40.88 105.69 ± 26.28 ^2,4^
		F (*n* = 36)	570.17 ± 115.09 ^4^				
	2	M (*n* = 74)	594.88 ± 150.42 ^5^	206.68 ± 54.51 201.38 ± 78.51 ^1,3^	73.52 ± 37.49 76.77 ± 39.61 ^1,3,4^	10.66 ± 9.59 ^4^ 6.55 ± 6.27 ^1,3^	84.18 ± 45.34 83.33 ± 44.13 ^1,3^
		F (*n* = 84)	538.20 ± 198.80 ^4,5^				
	3	M (*n* = 77)	557.12 ± 179.43 ^2^	218.92 ± 59.93 231.57 ± 56.74 ^2,4^	82.65 ± 38.08 92.68 ± 31.53 ^2^	12.47 ± 9.82 ^4^ 8.95 ± 6.38 ^2,4^	95.12 ± 46.51 101.63 ± 35.98 ^2,4^
		F (*n* = 71)	549.06 ± 160.89 ^4^				
	4	M (*n* = 68)	580.01 ± 225.52 ^5^	198.21 ± 79.67 203.51 ± 64.52 ^1,3,5^	66.77 ± 38.85 74.92 ± 34.71 ^1,3^	6.98 ± 6.97 ^1,2,3^ 5.55 ± 5.21 ^1,3^	73.76 ± 44.79 80.47 ± 38.95 ^1,3^
		F (*n* = 69)	632.72 ± 213.89 ^1,2,3,5^				
	5	M (*n* = 30)	443.90 ± 216.89 ^1,2,3,4^	190.69 ± 91.38	71.69 ± 46.12	9.77 ± 9.94	81.46 ± 54.79
		F (*n* = 29)	494.79 ± 155.67 ^2,4^	221.45 ± 76.60 ^4^	82.76 ± 37.83	7.04 ± 4.64	89.80 ± 41.66
Private school	1	M (*n* = 86)	650.37 ± 214.46 ^2,3,4^	199.48 ± 63.22 ^2^	65.90 ± 33.83 ^2,3^	8.04 ± 7.63 ^2,4^	73.94 ± 40.03 ^2,3,4^
		F (*n* = 86)	633.40 ± 187.95 ^2,3,4^	205.29 ± 58.18 ^2^	72.64 ± 32.55 ^2,4^	5.38 ± 4.64	78.03 ± 36.05 ^2,4^
	2	M (*n* = 39)	532.49 ± 125.95 ^1,5^	231.31 ± 50.33 ^1,3^	95.32 ± 30.21 ^1,3,5^	13.49 ± 8.65 ^1,3,4,5^	108.81 ± 36.98 ^1,3,4,5^
		F (*n* = 36)	549.45 ± 76.56 ^1,5^	249.94 ± 37.28 ^1,3,4,5^	92.33 ± 25.78 ^1,3,5^	7.70 ± 4.08	100.03 ± 28.25 ^1,3,5^
	3	M (*n* = 85)	546.16 ± 172.96 ^1,5^	207.13 ± 71.73 ^2^	78.85 ± 42.58 ^1,2^	9.76 ± 8.62 ^2^	88.61 ± 49.83 ^1,2^
		F (*n* = 76)	535.72 ± 173.64 ^1,5^	215.33 ± 67.97 ^2^	79.29 ± 34.15 ^2^	6.49 ± 5.69	85.79 ± 38.872
	4	M (*n* = 37)	542.29 ± 192.60 ^1,5^	210.88 ± 71.40	81.14 ± 43.78	13.06 ± 11.65 ^1,5^	94.20 ± 54.11 ^1^
		F (*n* = 42)	548.16 ± 161.27 ^1,5^	222.39 ± 57.21 ^2^	87.88 ± 39.35 ^1^	8.53 ± 8.75	96.41 ± 46.09 ^1^
	5	M (*n* = 36)	699.04 ± 162.36 ^2,3,4^	210.00 ± 55.11	68.40 ± 28.70 ^2^	6.74 ± 5.81 ^2,4^	75.15 ± 33.47 ^2^
		F (*n* = 44)	608.17 ± 217.07 ^2,3,4^	199.65 ± 79.12 ^2^	73.91 ± 40.62 ^2^	6.70 ± 5.55	80.62 ± 45.20 ^2^

Data are presented as mean ± SD. M: male; F: female; WASO: wake after sleep onset. ^1,2,3,4,5^ Statistical inter group differences between SEPs.

**Table 9 children-10-00551-t009:** Correlation of the daily sleep quality and PA analysis.

	Medea	BMI (kg/m^2^)	Sleep Efficiency (%)	Total Bed Time (min)	Total Sleep Time (min)	WASO (min)	Sedentary	Light PA (min)	Moderate PA (min)	Vigorous PA (min)	MVPA (min)
Medea	1										
BMI (kg/m^2^)	0.00	1									
Sleep Efficiency (%)	0.00	0.15 **	1								
Total bed time (min)	0.02	−0.16 **	−0.38 **	1							
Total sleep time (min)	0.02	−0.12 **	0.00	0.90 **	1						
WASO (min)	−0.00	−0.16 **	−0.81 **	0.53 **	0.22 **	1					
Sedentary	−0.04	0.22 **	0.02	−0.16 **	−0.16 **	−0.06 *	1				
Light PA (min)	0.01	−0.30 **	−0.24 **	0.21 **	0.13 **	0.27 **	−0.30 **	1			
Moderate PA (min)	0.00	−0.39 **	−0.23 **	0.25 **	0.19 **	0.26 **	−0.38 **	0.82 **	1		
Vigorous PA (min)	−0.02	−0.40 **	−0.20 **	0.26 **	0.21 **	0.24 **	−0.30 **	0.59 **	0.80 **	1	
MVPA (min)	−0.00	−0.41 **	−0.23 **	0.26 **	0.20 **	0.26 **	−0.39 **	0.80 **	0.99 **	0.86 **	1

Statistical significance in a correlation. * *p* ≤ 0.05; ** *p* ≤ 0.01.

## Data Availability

Data supporting reported results can be found by mailing authors.

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
