# Peer review of "Effects of Socioeconomic Environment on Physical Activity Levels and Sleep Quality in Basque Schoolchildren"

_children, 2023, doi:10.3390/children10030551_

Round 1

Reviewer 1 Report

The authors analyzed the relationship between physical activity and sleep and the socio-economic level 22 of children in the Basque Country. There, findings showed a positive trend in both sedentary behavior and sleep efficiency in Basque schoolchildren with an intermediate SEP with respect to the extremes. In addition, it was observed that females performed less high-intensity PA than males in all SEP groups. Also, they showed that more sedentary behavior was associated with less time spent in bed and poorer sleep efficiency. Similarly, more PA, regardless of its intensity, was associated with better body composition. Thus, in general, the present study is more interesting to address future strategies in a cross-cutting manner so that schoolchildren extend their hours of movement both in and out of the classroom. The article is well written.  The introduction and methodology are supporting the proposed aim. the following requests might be addressed

1-       The authors might have the ability to do the sample power calculation if all participants completed the study at the end or if there was any drop-out in the data collection process.  As mentioned in the limitations participants were requested to remove the accelerometer from the wrist in the case of aquatic activities and baths or showers.

2-       The author could add a flow chart of the study steps (if available)

Author Response

Dear reviewer,

On behalf of my co-authors, I would like to thank you for the opportunity to revise and resubmit our manuscript children-2269516, entitled “Effects of socio-economic environment on physical activity levels and sleep quality in Basque schoolchildren”. We found the reviewers’ comments to be helpful in revising the manuscript and have carefully considered and responded to each suggestion.

We have included a response to reviewer in which we address each comment the reviewer made. In our response, the comments are numbered and indicate the line or group of lines in which they are found in the manuscript.

Thank you again for your consideration of our revised manuscript.

Sincerely,

Arkaitz Larrinaga

Reviewer 2 Report

First of all, I would like to thank the authors of the manuscript for the effort they have put into the preparation of the manuscript entitled: Effects of socio-economic environment on physical activity levels and sleep quality in Basque schoolchildren. The study was aimed to analyse the relationship between PA and sleep with the socio-economic level of girls and boys between 6 and 17 years of age in the Basque Country.

I would be grateful if the authors would consider the following comments derived from my review of the manuscript:

1) I would like to congratulate the authors of the manuscript for the quality of the "introduction" section. The importance of physical activity and the factors that determine it have been pertinently addressed. The introduction needs to be expanded by referring to the time and quality of sleep in young people, as well as the factors that condition it. In addition, the relationship between physical activity and sleep should also be discussed in more depth. Some references that may be useful are listed below:

Sanz-Martín, D.; Zurita-Ortega, F.; Ruiz-Tendero, G.; Ubago-Jiménez, J.L. Moderate–Vigorous Physical Activity, Screen Time and Sleep Time Profiles: A Cluster Analysis in Spanish Adolescents. Int. J. Environ. Res. Public Health 202320, 2004.

Xu, F.; Adams, S.K.; Cohen, S.A.; Earp, J.E.; Greaney, M.L. Relationship between Physical Activity, Screen Time, and Sleep Quantity and Quality in US Adolescents Aged 16–19. Int. J. Environ. Res. Public Health 201916, 1524

Hrafnkelsdottir, S.M.; Brychta, R.J.; Rognvaldsdottir, V.; Chen, K.Y.; Johannsson, E.; Gudmundsdottir, S.L.; Arngrimsson, S.A. Less screen time and more physical activity is associated with more stable sleep patterns among Icelandic adolescents. Sleep Health 20206, 609–617

Gariepy, G.; Danna, S.; Gobiņa, I.; Rasmussen, M.; Gaspar, M.; Tynjälä, J.; Janssen, I.; Kalman, M.; Villeruša, A.; Husarova, D.; et al. How Are Adolescents Sleeping? Adolescent Sleep Patterns and Sociodemographic Differences in 24 European and North American Countries. J. Adolesc. Health. 202066, S81–S88.

Sanz-Martín, D.; Ubago-Jiménez, J.L.; Ruiz-Tendero, G.; Zurita-Ortega, F.; Melguizo-Ibáñez, E.; Puertas-Molero, P. The Relationships between Physical Activity, Screen Time and Sleep Time According to the Adolescents’ Sex and the Day of the Week. Healthcare 202210, 1955

Lee, P.H.; Tse, A.C.Y.; Wu, C.S.T.; Mak, Y.W.; Lee, U. Temporal association between objectively measured smartphone usage, sleep quality and physical activity among Chinese adolescents and young adults. J. Sleep Res. 202130, e13213

Ghekiere, A.; Van Cauwenberg, J.; Vandendriessche, A.; Inchley, J.; Gaspar de Matos, M.; Borraccino, A.; Gobina, I.; Tynjälä, J.; Deforche, B.; De Clercq, B. Trends in sleeping difficulties among European adolescents: Are these associated with physical inactivity and excessive screen time? Int. J. Public Health 201964, 487–498

Sanz-Martín, D.; Melguizo-Ibáñez, E.; Ruiz-Tendero, G.; Zurita-Ortega, F.; Ubago-Jiménez, J.L. Physical Activity, Energy Expenditure, Screen Time and Social Support in Spanish Adolescents—Towards an Explanatory Model about Health Risk Factors. Int. J. Environ. Res. Public Health 202219, 10222.

2) The abbreviation VVMFPA appears on line 154. It is necessary to write the full name of what is being referred to before putting the abbreviation. Check if the abbreviation is correct because it may refer to MVPA.

3) In the "subjects and design" section, specify how the physical activity has been differentiated according to its intensity. For example, specify the number of accelerations and/or the reference consulted in this regard.

4) In line 195 a first person plural personal pronoun (we) has been used. In other lines the possessive determinant "our" is also used. In scientific texts the third person singular/plural or indirect style should be used.

5) Table 2 should include a column specifying the average per capita income of each group.

6) The paragraph from lines 202-212 is repeated in the "procedure" section. I believe that the text on lines 202-212 should be deleted.

7) “Statistical analysis" section. It should be specified which post-hoc test has been used as a complement to the Kruskal-Wallis test to know between which groups there are significant differences. For example, Tukey's contrast analysis.

8) In the results section, the values obtained from the Mann-Whitney U test and the Kruskal-Wallis one-factor test should be specified. The information can be included in the tables or as a redacted text.

9) In the results section or, as supplementary material to the manuscript, the values obtained in the post-hoc test complementary to the Kruskal-Wallis test should be included.

In the first reference cited in the first comment of this review, an example of how to improve the manuscript can be seen by referring to comments 8 and 9.

10) Review the notes to tables 6 and 8. There are abbreviations in the notes that have not been used in the tables.

11) Tables in the results section. There are numbers that appear with three decimal places. Round to two decimal places.

12) At the beginning of the "discussion" section, the research objective should be included.

13) Line 404 mentions that participants had to fill in a personal diary. This instrument should be specified in section 2.2.

14) At the end of the discussion section, some future lines of research related to the subject of the manuscript should be included.

I would like the authors of the manuscript to take into consideration all the comments made after the revision of the manuscript.

Author Response

(The authors gave the same response as above.)

Round 2

Reviewer 2 Report

Dear Authors,

I would like to thank you for taking into consideration all the comments I made in the previous review. Undoubtedly, the quality of the manuscript has improved substantially. 

I consider that no further modifications are necessary for the article to be published.

Kind regards